# Tonic Activation of Extrasynaptic NMDA Receptors Decreases Intrinsic Excitability and Promotes Bistability in a Model of Neuronal Activity

**DOI:** 10.3390/ijms21010206

**Published:** 2019-12-27

**Authors:** David Gall, Geneviève Dupont

**Affiliations:** 1Laboratoire de Physiologie et Pharmacologie (CP604), Faculté de Médecine, Université Libre de Bruxelles, Route de Lennik 808, B-1070 Bruxelles, Belgium; 2Unité de Chronobiologie Théorique (CP231), Faculté des Sciences, Université Libre de Bruxelles, Boulevard du Triomphe, B-1050 Bruxelles, Belgium; gdupont@ulb.ac.be

**Keywords:** calcium, neuron, NMDA receptor, excitability, bistability, memory, Alzheimer’s disease, cerebral ischemia

## Abstract

NMDA receptors (NMDA-R) typically contribute to excitatory synaptic transmission in the central nervous system. While calcium influx through NMDA-R plays a critical role in synaptic plasticity, experimental evidence indicates that NMDAR-mediated calcium influx also modifies neuronal excitability through the activation of calcium-activated potassium channels. This mechanism has not yet been studied theoretically. Our theoretical model provides a simple description of neuronal electrical activity that takes into account the tonic activity of extrasynaptic NMDA receptors and a cytosolic calcium compartment. We show that calcium influx mediated by the tonic activity of NMDA-R can be coupled directly to the activation of calcium-activated potassium channels, resulting in an overall inhibitory effect on neuronal excitability. Furthermore, the presence of tonic NMDA-R activity promotes bistability in electrical activity by dramatically increasing the stimulus interval where both a stable steady state and repetitive firing can coexist. These results could provide an intrinsic mechanism for the constitution of memory traces in neuronal circuits. They also shed light on the way by which β-amyloids can alter neuronal activity when interfering with NMDA-R in Alzheimer’s disease and cerebral ischemia.

## 1. Introduction

N-methyl D-aspartate receptors (NMDA-R) are glutamate-gated ion channels and have long been of major interest to neuroscientists owing to their multiple implications in normal neuronal physiology. Transient activation of NMDA-R at the synaptic level is well known for its key role in the long-term potentiation (LTP) and the long-term depression (LTD) of synaptic transmission, which are believed to be the cellular correlate of learning at the synaptic level [1,2,3,4]. In addition to this phasic activity at the synapse, NMDA-R can also be tonically active [5] and the receptors mediating this tonic conductance are probably of extra-synaptic origin [6,7]. Besides, NMDA-R is also a key player in neuropathological situations like Alzheimer’s disease (AD) and cerebral ischemia. The respective influences of synaptic and extrasynaptic NMDA receptors on neuronal excitability are much investigated in AD. Several studies indeed suggest that the balance between synaptic and extrasynaptic NMDA-R is disturbed in AD patients [8,9]. Cerebral ischemia leads to a rise in extracellular glutamate [10]. In this condition, NMDA-R also appear to play a major role in neuronal calcium influx as the pharmacological blockade of these receptors reduces calcium entry and has a neuroprotective effect [11].

NMDA-R are generally considered as mediating an excitatory effect on neuronal activity. However conflicting results have been reported, as the pharmacological blockade of NMDA-R decreases the excitability of hippocampal neurons [5] but leads to hyperexcitabilty in medium spiny neurons (MSNs) [12], the main projection neurons in the striatum. This hyperexcitability is also observed when NMDA-R are genetically impaired in MSNs [13] or in primary sensory neurons of the dorsal root ganglions [14]. These apparently contradictory findings may be related to the possible coupling of these receptors with Ca${}^{2+}$ activated K${}^{+}$ channels. NMDA-R are indeed highly permeable to calcium and coupling between calcium influx through NMDA-R and Ca${}^{2+}$ activated K${}^{+}$ channels has been demonstrated to lead to changes in neurotransmission [15,16] and intrinsic excitability [12,14].

In this paper, we use mathematical modeling to explore the consequences of the tonic activation of extrasynaptic NMDA receptors on the control of neuronal excitability by Ca${}^{2+}$ activated K${}^{+}$ channels. This theoretical approach allows a better understanding of the underlying mechanisms, which is difficult to obtain experimentally. We show that tonic activation of NMDA-R is able to reduce intrinsic neuronal excitability during spike generation and to dramatically increase the stimulus interval where a regime of sustained electrical activity coexists with a stable resting state. The latter effect could provide a mechanism allowing the encoding of memory traces in neuronal circuits by persistent changes in intrinsic firing properties. The coexistence between an active and a resting state could also have a significance in the framework of AD and cerebral ischemia. Indeed, this coexistence highlights the intimate relation between calcium dishomeostasis and altered neuronal activity, and could provide an explanation for the observed perturbations in memory formation.

## 2. Results

### 2.1. Theoretical Model

We propose a simple theoretical model that describes neuronal electrical activity, including the tonic activity of NMDA receptors and a cytosolic calcium compartment (Figure 1A). This model should be seen as a minimal model allowing the qualitative study of the impact of the coupling between tonic NMDA activity and Ca${}^{2+}$ activated K${}^{+}$ channels on the oscillatory regimes of an excitable cell. Nevertheless, the core of this model has been experimentally validated on cerebellar granule cells, as it correctly predicts hyperexcitability, when the cytosolic calcium buffering capacity is decreased by by a null mutation suppressing the endogenous calretinin expression, [17] and the transition from regular spiking to bursting, when the calcium buffering capacity of the cytosol is increased by injection of an exogenous buffer [18]. These results already demonstrated that the activation of Ca${}^{2+}$ activated K${}^{+}$ channels can provide a tight coupling between excitability and calcium dynamics during spike generation. We extend this model to take tonic NMDA-R into account, as our purpose is to study how the presence of this current alters neuronal activity. The model considers five variables and a single compartment. The membrane potential dynamics are governed by the current balance equation:(1)$$\begin{array}{ccc} C_{m}\frac{dV}{dt} & = & -I_{\mathrm{Na}(V)}-I_{K(V)}-I_{\mathrm{Ca}(V)}-I_{K(\mathrm{Ca})}-I_{NMDA}+I_{inj} \end{array}$$
where $C_{m}$ is cell capacitance, $I_{\mathrm{Na}}$ is a voltage-dependent ${\mathrm{Na}}^{+}$ current, $I_{K(V)}$ a delayed rectifier $K^{+}$ current, $I_{\mathrm{Ca}}$ a high-threshold voltage dependent ${\mathrm{Ca}}^{2+}$ current and $I_{K-\mathrm{Ca}}$ a ${\mathrm{Ca}}^{2+}$-activated $K^{+}$ current. These ionic currents have been shown to be at the core of action potential generation in cerebellar granule cells [19]. The equations governing the evolution of the gating variables for these different ionic currents and all parameters values are the same as used previously [17,18] and can be found in the Appendix A.

An additional term, $I_{NMDA}$, has been included in order to take into account the existence of tonically active NMDA receptors (Figure 1B) and has been adapted from a previous work [20]. Due to the ionic selectivity of these receptors, the total NMDA current is the sum of three components related to the flow of Na${}^{+}$, K${}^{+}$, and Ca${}^{2+}$ ions:(2)$$\begin{array}{ccc} I_{NMDA} & = & I_{NMDA,\mathrm{Na}}+I_{NMDA,K}+I_{NMDA,\mathrm{Ca}} \end{array}$$
Each component of the tonically active current is described by the Goldman–Hodgkin–Katz current densities multiplied by A, the cell area: (3)$$\begin{array}{ccc} I_{NMDA,\mathrm{Na}} & = & A\;{\bar{P}}_{NMDA}\;P_{\mathrm{Na}}\;MgB\left(V\right)\;\frac{VF^{2}}{RT}\frac{{\left[\mathrm{Na}\right]}_{i}-{\left[\mathrm{Na}\right]}_{o}e^{-FV/RT}}{1-e^{-FV/RT}} \end{array}$$(4)$$\begin{array}{ccc} I_{NMDA,K} & = & A\;{\bar{P}}_{NMDA}\;P_{K}\;MgB\left(V\right)\;\frac{VF^{2}}{RT}\frac{{\left[K\right]}_{i}-{\left[K\right]}_{o}e^{-FV/RT}}{1-e^{-FV/RT}} \end{array}$$(5)$$\begin{array}{ccc} I_{NMDA,\mathrm{Ca}} & = & A\;{\bar{P}}_{NMDA}\;P_{\mathrm{Ca}}\;MgB\left(V\right)\;\frac{4VF^{2}}{RT}\frac{{\left[\mathrm{Ca}\right]}_{i}-{\left[\mathrm{Ca}\right]}_{o}e^{-2FV/RT}}{1-e^{-2FV/RT}} \end{array}$$
where ${\bar{P}}_{NMDA}$ is the maximum permeability of the NMDA-R, which has a value of 6.37 nm/s, close to the value of 10 nm/s used in the original model [20]. This value gives a realistic NMDA current intensity (Figure 1B) for a cerebellar granule cell [21]. The permeability ratio of the NMDA-R to the different ions, $P_{\mathrm{Ca}}:P_{K}:P_{\mathrm{Na}}$, is set to 10.6:1:1 [22]. Ionic concentrations are ${\left[\mathrm{Na}\right]}_{i}$ = 18 mM, ${\left[\mathrm{Na}\right]}_{o}$ = 140 mM, ${\left[K\right]}_{i}$ = 140 mM, ${\left[K\right]}_{o}$ = 5 mM, ${\left[\mathrm{Ca}\right]}_{i}$ = 100 nM and ${\left[\mathrm{Ca}\right]}_{o}$ = 2 mM. The temperature is *T* = 35 ${}^{\circ }$C, *F* is the Faraday constant and *R* is the gas constant. The MgB(V) function governs the voltage dependency of the magnesium block of the NMDA current and is given by:(6)$$\begin{array}{ccc} MgB(V) & = & {\left\{1+\frac{{\left[\mathrm{Mg}\right]}_{o}e^{-0.062V}}{3.57}\right\}}^{-1} \end{array}$$
where ${\left[\mathrm{Mg}\right]}_{o}$ = 2 mM [23].

Finally, the constant term $I_{inj}$ corresponds to the current injected into the cell to study its excitability properties. Intrinsic excitability is evaluated by measuring the voltage response while injecting steps of depolarizing current of increasing intensities in the presence or in the absence of NMDA-R activity. The slopes of the almost linear part of the resulting current frequency plots were used as measures of excitability.

The evolution of the free calcium concentration (in μM) is given by the following balance equation, adapted from the original cerebellar granule cell model [17], including a term for the Ca${}^{2+}$ flux through the NMDA-R:(7)$$\begin{array}{cc} \frac{d\mathrm{Ca}}{dt} & =f\left[-\frac{I_{\mathrm{Ca}(V)}+qI_{NMDA,\mathrm{Ca}}}{2FV_{shell}}-\beta_{\mathrm{Ca}}\mathrm{Ca}\right] \end{array}$$
where the dimensionless parameter *f* represents the calcium buffering capacity of the cytosol, *q* is scaling factor controlling the calcium entry through NMDA-R (which is equal to 1 unless stated otherwise), $V_{shell}$ is the volume of the cytosolic compartment, and $\beta_{\mathrm{Ca}}$ is the rate of Ca${}^{2+}$ removal from the cytosol (see Appendix A for parameter values).

### 2.2. Tonic NMDA-R Activity Decreases Neuronal Activity

The effect of tonic NMDA-R activity on neuronal electrical activity has been evaluated by computing the membrane potential while injecting steps of depolarizing current of increasing intensities, between 0 to 30 pA, as this corresponds to a realistic stimulus range for cerebellar granule cells [19]. Numerical simulations were performed in the absence and presence of tonic NMDA-R activity. In the absence of tonic NMDA-R activity, periodic action potentials occur above a threshold value of the injected current equal to 2 pA (Figure 2A, left panel). The frequency of action potentials increases with the amplitude of the injected current (Figure 2B, black curve). In the presence of tonic NMDA-R activity, the action potentials frequency is decreased at 25 pA injected current and there is no electrical activity at 2 pA (Figure 2A, right panel). A minimal current of 12 pA is indeed required to generate action potentials. Moreover, the current-frequency plot (Figure 2B, red curve) shows a significant decrease in slope, from 1.55 Hz/pA in the absence of tonic NMDA-R activity to 1.13 Hz/pA in the presence of such an activity. Together with the increase of the value of the firing threshold, this clearly indicates a decrease in neuronal activity. This is a counterintuitive result as NMDA-R are usually known to exert an excitatory effect.

To test our hypothesis that this inhibitory effect is mediated by the calcium entry through the NMDA-R, we have examined the electrical response of the neuron in the presence or absence of the calcium flux through the NMDA-R. This is achieved by suppressing the $I_{NMDA,Ca}$ term only in the balance equation governing the calcium dynamics (Equation (Equation 7) with q=0), leaving the contribution of NMDA-R unchanged in the current balance equation (Equation (Equation 1)). In a cellular context, this would correspond to a situation where the calcium entering through the NMDA-R would have a negligible impact on the calcium concentration in the vicinity of the ${\mathrm{Ca}}^{2+}$-activated $K^{+}$ channels because the two channels would not be localized in the same nanodomain. This is plausible because Ca${}^{2+}$ is highly buffered in the cytoplasm [24]. Therefore, the effective distance between calcium influx through NMDA-R and the calcium sensor of the ${\mathrm{Ca}}^{2+}$-activated $K^{+}$ channels would prevent the coupling between the two channels through calcium changes, but the current flowing through the NMDA-R would still contribute to the overall excitability of the cell. In this situation, we see that the NMDA-R indeed exert an excitatory effect, as the neuron is firing even in the absence of injected current. Additionally, for a given value of injected current, the frequency is larger in the absence of coupling between NMDA-R activity and ${\mathrm{Ca}}^{2+}$-activated $K^{+}$ channels (dashed line in Figure 2C).

### 2.3. Activation of Calcium Activated Potassium Channels Couples Excitability and Tonic NMDA-R Activity

To understand the observed changes in neuronal firing in the presence of tonic NMDA-R activity, we have investigated the effect of calcium influx through the NMDA receptors on the activation of the ${\mathrm{Ca}}^{2+}$-activated $K^{+}$ current. Figure 3A shows intracellular calcium concentration and ${\mathrm{Ca}}^{2+}$-activated $K^{+}$ current time courses, at a suprathreshold injected current intensity of 25 pA, in the absence or presence of tonic NMDA-R activity. The flux of calcium ions through the NMDA-R has a modest effect on the maximal amplitude of the calcium transients but substantially increases the residual calcium concentration between the spikes. This increased residual calcium concentration leads to a higher activation of the ${\mathrm{Ca}}^{2+}$-activated $K^{+}$ channels during the interspike interval and the action potential initiation (Figure 3B). The corresponding persistent hyperpolarizing current is therefore responsible for the increased spiking threshold current and the reduction of the firing frequency in the presence of tonic NMDA activity.

### 2.4. Tonic NMDA-R Activity Induces Bistability in Neuronal Firing

We used bifurcation analysis to investigate how the presence of a tonic NMDA-R activity influences the dynamics of the neuronal activity. In both cases, without or with tonic NMDA-R activity, after a threshold value of injected current, the steady state loses its stability at a Hopf bifurcation point (Figure 4). In addition, in both cases, the bifurcation is subcritical. Thus, for a range of values of the injected current, a stable steady state coexists with a stable limit cycle (oscillations). When increasing the injected current progressively, at the subcritical bifurcation point, the stable stationary point becomes unstable, and the trajectory spirals out to find a pre-existing limit cycle, jumping to a finite firing frequency. As a consequence of the subcritical character of the bifurcation, neuronal activity can display a bistable behavior, since a stable steady state (resting membrane potential) coexists with a stable limit cycle (repetitive action potentials) in a range of values of injected current. Our results demonstrate that tonic NMDA-R activity dramatically increases the injected current interval where such a bistable behavior is present (shaded area on Figure 4), from 1 pA in the absence of tonic NMDA-R activity to more than 10 pA when the receptors are active. This is in line with several previous studies showing that bistability between repetitive firing and quiescence can be obtained when voltage-dependent calcium entry competes with calcium sensitive potassium conductance responsible for after spike hyperpolarization [25,26,27].

Figure 4 shows that the presence of tonic NMDA-R does not much affect the characteristics of repetitive firing, but largely extends the range of stability of the steady state. This suggests that the activation of the ${\mathrm{Ca}}^{2+}$-activated $K^{+}$ channels by calcium flowing through the NMDA-R may be responsible for this effect. Indeed, when the cell is quiescent, an increase in the ${\mathrm{Ca}}^{2+}$-activated $K^{+}$ current, due to an in increase in cytosolic calcium, hyperpolarizes the cell membrane. This stabilizes the steady state for larger values of injected current. However, as the ${\mathrm{Ca}}^{2+}$-activated $K^{+}$ current does not significantly modify the action potential (see Figure 3B), it does hardly alter the stability of the repetitive firing. Thus, both regimes (stable steady-state and repetitive firing) coexist on an extended range of applied current. To validate this hypothesis about the origin of bistability, we checked that its range of occurrence is also modified by NMDA-R independent processes affecting intracellular calcium. As shown in Table 1, an increase in the conductance of voltage gated ${\mathrm{Ca}}^{2+}$ channels ($g_{\mathrm{Ca}}$), a decrease in the rate of calcium pumping ($\beta_{\mathrm{Ca}}$), and a decrease in the cell calcium buffering capacity (reflected by an increase in the parameter *f*) all three favor the occurrence of bistability. In any case, the effect is reduced if the conductance of the ${\mathrm{Ca}}^{2+}$-activated $K^{+}$ ($g_{K-\mathrm{Ca}}$) is decreased.

### 2.5. Tonic NMDA-R Activity Increases Hysteresis in Neuronal Firing

Such increase in the bistability domain, where either a limit cycle or a stable steady state can be present, promotes hysteresis in the neuronal response as the firing threshold will be more sensitive to the history of stimulation. This effect can be clearly demonstrated if we submit the neuron to increasing and decreasing ramps of stimuli, in absence or presence of tonic NMDA-R activity. The difference in the current intensity for the onset and offset of the firing is dramatically increased when the NMDA-R are tonically active (Figure 5A). The effect is so prevalent that, in the presence of NMDA-R activity, the neuron submitted to a decreasing ramp stimulus keeps firing even when stimulus intensity drops to 0 pA. Furthermore, in the presence of tonic NMDA-R activity, a transient depolarizing current locks neuronal activity in a continuous firing mode. This memory effect disappears in the absence of tonic NMDA-R activation (Figure 5B).

## 3. Discussion

This study demonstrates that tonic activation of NMDA-R can lead to an overall decrease in excitability. The current intensity necessary to induce spiking is indeed higher. Moreover, when the injected current exceeds the threshold, the firing frequency is lower in the presence than in the absence of tonic NMDA-R activity. This effect is mediated by the calcium influx through NMDA-R, which increases the residual free calcium concentration between the spikes and therefore increases the activation of calcium activated potassium channels. Such an activation occurs when electrical activity and calcium dynamics are coupled strongly during the generation of the spike, through the activation of ${\mathrm{Ca}}^{2+}$-activated $K^{+}$ channels. If coupling is weak, as for example when these channels and NMDA-R are too far apart to allow calcium dependent activation to occur, tonic NMDA-R activity will lead to an increase in the overall excitability. This may explain the conflicting experimental results reported in the literature regarding the effect of tonic NDMA current on intrinsic excitability in different neuronal populations [5,12,13,14].

In addition, we show that the tonic NMDA-R activation promotes bistability in the neuronal activity. For a range of stimulus intensities, a stable steady-state solution co-exist with a stable limit cycle of repetitive firing. The range of stimulating current where these two behaviors co-exist is dramatically increased in the presence of tonic NMDA-R activity. Interestingly, such a quiescence/firing bistability has been shown to occur in a motoneuron model [25] when there is competition between a voltage-dependent calcium entry, L-type calcium channels, and ${\mathrm{Ca}}^{2+}$-activated $K^{+}$ channels at the somatic level. Our results show that calcium entry through tonically active NDMA-R provides another mechanism leading to the same type of bistability. Functionally, such a bistable behavior allows the encoding of a transient signal [28], like a brief synaptic event, by a long-lasting change in firing. Indeed, in the presence of tonic activation of extrasynaptic NMDA-R, if an initially silent cell is activated by a transient depolarizing current of synaptic origin, the neural activity will be locked in a firing mode, as the system will stay on the limit cycle even after the disappearance of the stimulus. On the contrary, when submitted to a similar transient depolarizing current, a neuron lacking tonic NMDA-R activity will go back to a silent state when the stimulus intensity drops to zero.

At the neuronal circuit level, such changes in intrinsic neuronal properties could therefore provide a mechanism modifying circuit dynamics on the long term. Bistability endows neurons with rich forms of information processing and this property has already been hypothesized to be involved in memory formation [29]. Experimentally, intrinsic activation of dormant neurons, and their engram integration, have been shown to constitute a memory trace in cortical circuits [30,31]. The mechanism we propose here, in which tonic NMDA-R activity leads to bistability, could permit engram integration by providing long lasting adjustment of intrinsic firing properties, allowing to activate dormant or highly unreliable neurons into reliable responders and integrate them into functional neural ensembles. Ever since its discovery, long-term potentiation of synaptic transmission has been seen as the main cellular mechanism underlying information storage in the central nervous system. In a rather speculative manner, our theoretical results raise the possibility that extrasynaptic modulation of intrinsic neuronal excitability can happen under some condition. Therefore, synaptic plasticity may not be the sole mechanism of memory formation, a view that is supported by recent experimental results [32].

Finally, our results may also be relevant to understand the changes of neuronal activity induced by the presence of β-amyloids in the pathological contexts of Alzheimer’s disease and cerebral ischemia. According to the calcium hypothesis of Alzheimer’s disease, a positive circuit whereby intracellular calcium stimulates β-amyloids formation, which themselves tend to increase cytosolic calcium concentration plays a key role in the onset of the disease [33]. It is therefore interesting to investigate the relation between modest increases in intracellular calcium and neuronal activity. NMDA-R play an important role in this relation, as they are directly activated by β-amyloids oligomers [34]. Moreover, β-amyloids provoke the release of glutamate in the extracellular medium and decrease glutamate uptake [35,36] resulting in glutamate spillover. As a result, extrasynaptic NMDA-R are likely to be stimulated in a tonic manner, as modelled in the simulations shown in this study. Thus, our simulations of tonic stimulation of NMDA-R can be related to the situation of a neuron in the presence of β-amyloids. Our results suggest that the calcium influx due to activation of NMDA-R by β-amyloids could be responsible both for a decreased neuronal excitability and major alterations in memory formation, in agreement with the calcium hypothesis of Alzheimer’s disease [37]. The bistable behavior predicted by the model could also provide an explanation to the observed hyperactive neurons near amyloid plaques in a mouse model of Alzheimer’s disease [38], as these neurons may be locked in a state of repetitive firing after a transient stimulation. Our results are also in agreement with the fact that memantine, an inhibitor of extrasynaptic NMDA-R, appears to be an efficient drug to block or reverse the deleterious actions of β-amyloids [9,39]. Although the calcium hypothesis of Alzheimer’s disease remains speculative, increasing evidence indicates the importance of β-amyloids generation in cerebral ischemia. Cerebral ischemia leads to dysregulation of Alzheimer related genes [40,41], ultimately increasing β-amyloids formation [42]. It is therefore very likely that the tonic activity of extrasynaptic NMDA-R is stimulated in these conditions inducing major changes in memory formation by the mechanism we describe here. This could participate in the long lasting alterations in spatial learning and memory observed after a transient brain ischemia [43].

It should be kept in mind, however, that other calcium fluxes are sensitive to β-amyloids. It has been shown recently in hippocampal neurons that the increase in cellular calcium resulting from all these interactions causes an increase in the $K^{+}$ current and hippocampal network firing inhibition [44]. Much work remains to be done to consider all fluxes in a unified modeling approach, which should moreover consider the existence of two populations of NMDA-R (synaptic and extrasynaptic) as well as spatial aspects. Nevertheless, our theoretical results suggest that the calcium-activated potassium channels could provide an interesting therapeutic target.

## 4. Conclusions

Using a theoretical approach, we have investigated the role of tonic activation of extrasynaptic NMDA receptors in the control of neuronal excitability by Ca${}^{2+}$ activated K${}^{+}$ channels. Our results demonstrate that tonic NMDA-R activity negatively regulates intrinsic neuronal activity and promotes bistability by dramatically increasing the stimulus interval where both a stable steady state and repetitive firing can exist. At the neuronal circuit level, the mechanism we propose, leading to increased bistability, could provide a form of non-synaptic plasticity, allowing engram integration and therefore play an important role in memory formation in the central nervous system. On the other hand, the alterations to the intrinsic neuronal excitability in the presence of tonic activation of NMDA-R could be relevant in the context of Alzheimer’s disease and cerebral ischemia, as β-amyloids oligomers have been reported to stimulate NMDA-R.

## 5. Materials and Methods

The equations of the model are numerically solved using Heun’s method implemented in GNU Fortran (https://gcc.gnu.org/fortran/). Original source code is available in the ModelDB database (https://senselab.med.yale.edu/modeldb/). The bifurcation diagrams are built with the software XPPAUT 8.0 (Free Software Foundation, Inc., Cambridge, MA, USA, http://www.math.pitt.edu/~bard/xpp/xpp.html).

## Figures and Tables

**Figure 1 ijms-21-00206-f001:**
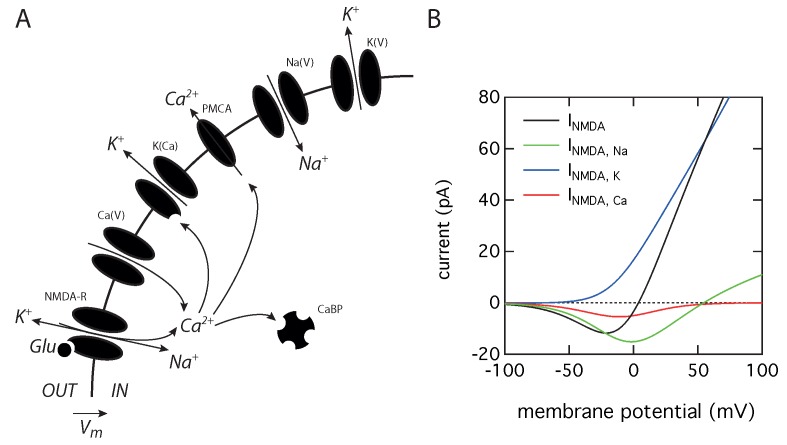
Model of neuronal activity including a tonically active NMDA-R current. (**A**) Schematic representation showing the various ligand- and voltage-gated channels used in the model: N-methyl-D-aspartate receptor (NMDA−R), high-voltage-gated calcium channels (Ca(V)), calcium-activated potassium channels (K(Ca)), plasma membrane calcium ATPase (PMCA), voltage-dependent sodium channels (Na(V)), and delayed rectifier potassium channels (K(V)). The model also takes into account intracellular calcium buffers (CaBP). (**B**) Current-voltage relationship for the three different ionic components ($I_{NMDA,\mathrm{Na}}$, $I_{NMDA,K}$, and $I_{NMDA,\mathrm{Ca}}$) and the total current ($I_{NMDA}$) flowing through the tonically activated NMDA receptors.

**Figure 2 ijms-21-00206-f002:**
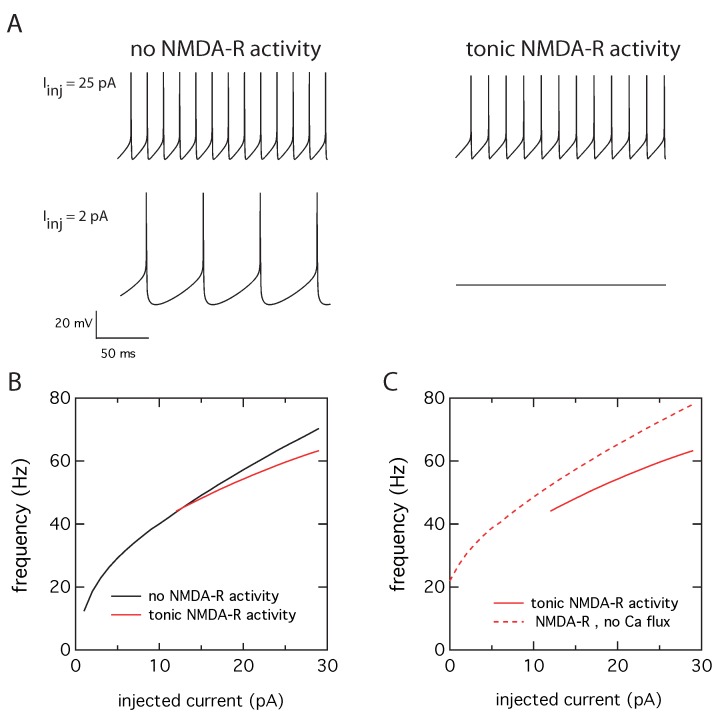
Decreased intrinsic excitability in presence of tonic NMDA-R activity. (**A**) The presence of tonic NDMA-R activity increases the threshold current for action potential generations and decreases the firing frequency for an injected current of 25 pA. (**B**) Corresponding current-frequency plots in absence (black) and presence of a tonic NMDA-R activity (red). (**C**) Hyperexcitability is observed when the $I_{NMDA,Ca}$ term is suppressed from the Ca${}^{2+}$ balance equation (dotted line), compared to the current-frequency curve obtained with a Ca${}^{2+}$ influx through the NMDA-R (red line, same as in (B)). The decreased intrinsic excitability in presence of tonic NMDA-R activity is therefore mediated by the Ca${}^{2+}$ influx through these receptors.

**Figure 3 ijms-21-00206-f003:**
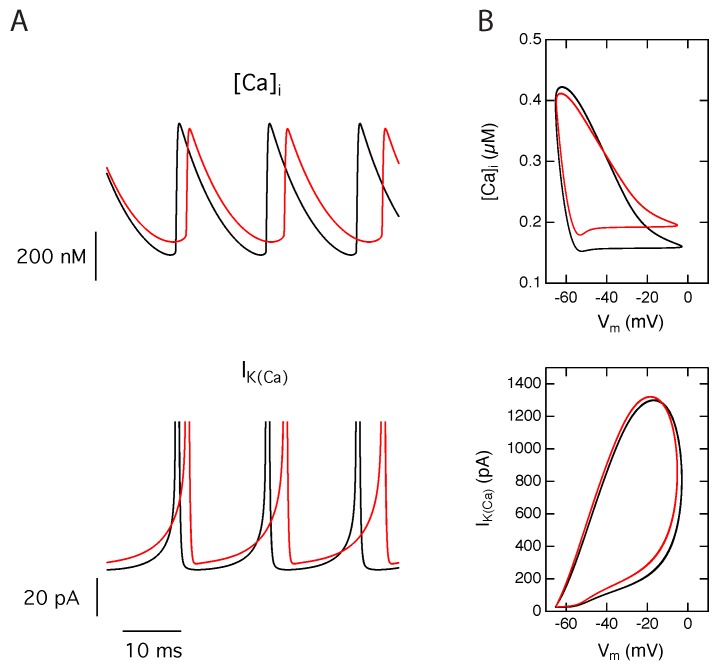
Tonic NMDA-R activity enhances calcium-activated potassium channels activation during periodic firing. (**A**) Time series for the intracellular calcium concentration (${\left[{\mathrm{Ca}}^{2+}\right]}_{i}$) and the calcium-activated potassium current ($I_{K(\mathrm{Ca})}$) in absence (black) or presence (red) of tonic NMDA-R activity, obtained for 25 pA injected current. Current traces are truncated to allow an expanded view of the low intensities. Between two action potentials, the residual calcium level is higher in presence of tonic NMDA-R activity. This results in a greater activation of calcium-activated potassium channels during the interspike interval and the action potential initiation, increasing the duration of the action potential afterhyperpolarization and therefore decreasing the firing frequency. (**B**) Corresponding trajectories in the $\left(V_{m},{\left[{\mathrm{Ca}}^{2+}\right]}_{i}\right)$ and $\left(V_{m},I_{K(\mathrm{Ca})}\right)$ planes show an increased residual calcium and increased activation of calcium-activated potassium channels.

**Figure 4 ijms-21-00206-f004:**
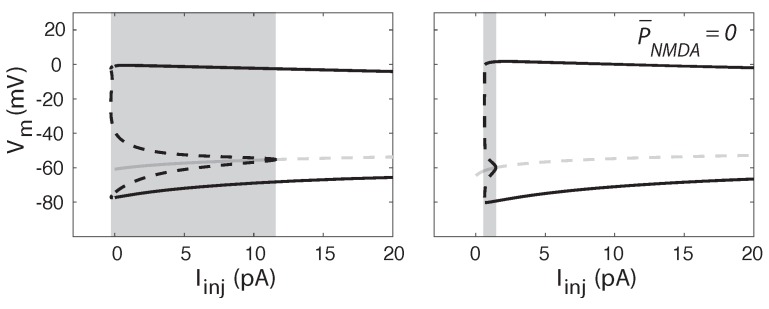
Bifurcation diagrams of the model in the presence (**left**) and absence (**right**) of tonic NMDA current. Gray and black lines indicate steady states and oscillatory solutions, respectively. In the latter case, the two branches represent the minimal and maximal values of voltage reached during oscillations. Plain lines represent stable solutions and dashed lines, unstable ones. In the presence of tonic NMDA activity, a stable state coexists with stable oscillations in an extended range of values for the injected current (shaded area). Sub-critical Hopf bifurcations occur at 11.5 pA (**left**) and 1.4 pA (**right**).

**Figure 5 ijms-21-00206-f005:**
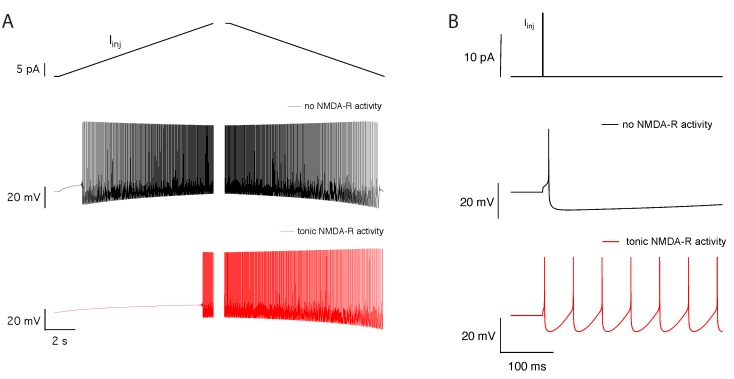
Tonic NMDA-R activity increases the hysteresis observed during a stimulation by a ramp or by a pulsed current injection. (**A**) The applied stimuli consist of a ramp of injected current from 0 pA to 20 pA or from 20 pA to 0 pA, during 10 s. The presented solutions correspond to a situation where the system starts from the stable steady-state at 0 pA or from the stable limit cycle at 20 pA. In absence of tonic NMDA-R activity, the threshold current for spike generation is nearly similar for increasing or decreasing ramp stimuli (black trace). In presence of tonic NMDA-R activity, the threshold current for spike generation greatly differs for increasing or decreasing ramp stimuli (red trace). Note that, in absence of tonic NMDA-R activity, the system goes back to the stable steady-state when submitted to the decreasing ramp stimulus but, when the receptors are active, it is still oscillating when it reaches 0 pA of injected current. (**B**) The applied stimuli consist of a single pulse of injected current from 0 pA to 15 pA, during 1 ms. In absence of tonic NMDA-R activity, a single spike is elicited (black trace). In presence of tonic NMDA-R activity, the short current pulse induces sustained spiking (red trace).

**Table 1 ijms-21-00206-t001:** Ranges of values of the injected current ($I_{inj}$) allowing bistability for different values of parameters affecting the concentration of cytosolic calcium. For each parameter, the middle column indicates the range obtained with the default value of the maximal conductance of the $I_{K-\mathrm{Ca}}$ current ($g_{K-\mathrm{Ca}}$ = 56.5 pS) and the right column indicates the range obtained with a reduced value of this maximal conductance ($g_{K-\mathrm{Ca}}$ = 50 pS). The values of the other parameters are listed in the Appendix A. Ranges of bistability are evaluated on the basis of bifurcation diagrams obtained as in Figure 4. Note that injected current intensities between 0 to 30 pA represent the realistic stimulus range for our model.

Changes VS. Default Values	$g_{K-\mathrm{Ca}}$ = 56.5 pS	$g_{K-\mathrm{Ca}}$ = 50 pS
−	0.81 pA	0.31 pA
$g_{\mathrm{Ca}}$ = 80 nS	3.71 pA	1.12 pA
$\beta_{\mathrm{Ca}}$ = 8 ms${}^{-1}$	4.01 pA	1.43 pA
*f* = 0.1	2.81 pA	1.51 pA

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
