# Peer review of "Tonic Activation of Extrasynaptic NMDA Receptors Decreases Intrinsic Excitability and Promotes Bistability in a Model of Neuronal Activity"

_ijms, 2019, doi:10.3390/ijms21010206_

Round 1

Reviewer 1 Report

This manuscript by Gall and Dupont describes s neuronal electrical activity including the tonic activity of eatrasynaptic NMDA receptors and a cytosolic calcium compartment by using mathematical modeling. The manuscript is well written.  English language and stile should be checked and revised.

Author Response

We are grateful to the reviewer #1 for his constructive remark. We have carefully checked the existing text to improve the quality of the manuscript; several sentences throughout the text were either re-written or removed.

Reviewer 2 Report

MS.664344. An interesting approach to the issue. I am not a mathematician but a medic, so my comments relate mainly to medical and editorial issues. Line 17 take out second long. Lines 21, 25 and 29 spaces. Line 30 what it is MSNs? Elements such as Ca and other and beta are not written in Italic. In the introduction, the omitted papers on cerebral ischemia should be added, where we have an identical change as in AD, Pluta R et al., Resuscitation. 1988 Jul;16(3):193-210, Salinska E et al., Exp Neurol. 1991 Apr;112(1):89-94. Lines 219 and 230 calcium hypothesis hardly exists, but ischemic AD theory is developing. Lines 224, 232 and 236 spaces. Line 240 what it is NMDAR? The discussion should include works related to ischemic AD theory, Pluta R et al., Anat Rec (Hoboken). 2009 Dec;292(12):1863-81, Kiryk A et al., Behav Brain Res. 2011 May 16;219(1):1-7, Kocki J et al., J Alzheimers Dis. 2015;47(4):1047-56, Pluta R et al., J Alzheimers Dis. 2016;51(4):1023-31,

Author Response

1) Editorial issues. We have carefully checked the existing text to improve the quality of the manuscript. All the editorial remarks from reviewer #2 have been taken into account and the manuscript has been modified accordingly. 

2) We thank the reviewer for this interesting suggestion that extends the implications of our results to neurological dysfunctions caused by cerebral ischemia, which is a very important cause of disability. We have therefore explicitly mentioned cerebral ischemia throughout the manuscript. More specifically, we have included the following paragraph and references in the introduction :

" Besides, NMDA-R is also a key player in neuropathological situations like Alzheimer's disease (AD) and cerebral ischemia. The respective influences of synaptic and extrasynaptic NMDA receptors on neuronal excitability are much investigated in AD. Several studies indeed suggest that the balance between synaptic and extrasynaptic NMDA-R is disturbed in AD patients (Huang et al., 2017; Rush et al., 2014). Cerebral ischemia leads to a rise in extracellular glutamate (Pluta et al.,1988). In this condition, NMDA-R also appear to play a major role in neuronal calcium influx as pharmacological blockade of these receptors reduces calcium entry and has a neuroprotective effect (Salinska et al., 1991). "

3) We have modified the discussion accordingly, by adding the following :

"Although the calcium hypothesis of Alzheimer's disease remains speculative, increasing evidence indicates the importance of beta-amyloids generation in cerebral ischemia. Cerebral ischemia leads to dysregulation of Alzheimer related genes (Kocki et al., 2015; Pluta et al., 2016), ultimately increasing beta-amyloids formation (Pluta et al., 2009). It is therefore very likely that the tonic activity of extrasynaptic NMDA-R is stimulated in these conditions inducing major changes in memory formation by the mechanism we describe here. This could participate in the long lasting alterations in spatial learning and memory observed after a transient brain ischemia (Kiryk et al. 2011)."

Round 2

Reviewer 2 Report

It is OK! Accepted.